# A Recursive Non-Uniform Sampling Estimator for Asynchronous Nonlinear Systems

**DOI:** 10.3390/s24092882

**Published:** 2024-04-30

**Authors:** Yu-Hang Yang, Jin-Gang Liu, Shen-Min Song

**Affiliations:** Center for Control Theory and Guidance Technology, Harbin Institute of Technology, Harbin 150001, China; 22b904066@stu.hit.edu.cn (J.-G.L.); songshenmin@hit.edu.cn (S.-M.S.)

**Keywords:** interpolation, modelling, state estimation, non-uniform sampling, covariance intersection fusion

## Abstract

In this paper, we consider the problem of asynchronous estimation in the presence of packet losses for the randomly sampling nonlinear system. Packet losses occur at the control input and at the measurement side. Firstly, the synchronization of the asynchronous sampling system is realized by weighting the state of the adjacent state update points. Secondly, the projection theorem is used to estimate the system state at the sampling time. Due to modeling errors and unmodeled dynamics, obtaining an accurate dynamic model is challenging. Therefore, observation inference based on interpolation techniques is proposed to solve the asynchronous estimation problem. Furthermore, the algorithm is extended to multi-sensor systems to obtain a distributed fusion estimator. Finally, simulation experiments are conducted to validate the effectiveness of the algorithm.

## 1. Introduction

Compared to the uniform sampling mechanism, the non-uniform sampling mechanism can better ensure the controllability and observability of the discretization system [1], which is widely used in battery health management [2], target tracking [3] and power systems [4]. Therefore, it is of great theoretical and practical significance to study the estimation problem of networked asynchronous sampling systems. The difficulty in solving this problem lies in synchronizing asynchronous sampling systems. After decades of development, the current synchronization methods for asynchronous non-uniform sampling systems are mainly divided into data block [5] and model transformation [6].

The introduction of data block theory provides an intuitive approach to asynchronous estimation problems. Representative work in this area is as follows. For the estimation problem of randomly sampling a neural network with an unknown sampling interval, this paper designs a density-identifiable estimator based on data reception [7]. The non-uniform sampling mechanism also poses a challenge for parameter identification in nonlinear systems. The decomposition technique effectively reduces the computational complexity of the algorithm [8]. In networked control systems, there are many uncertain factors, such as time delays, packet losses, fading measurements, random parameter uncertainty, correlated noise, and more. Using innovation analysis and the covariance intersection (CI) technique, an asynchronous fusion estimator that accounts for fading measurements is proposed [9]. Event-triggered mechanisms are crucial for achieving time synchronization in asynchronous sampling systems. A dynamic event-triggered mechanism [10] is used to develop the time-varying moving horizon asynchronous estimator. An event-triggered estimator is developed by incorporating packet losses and heavy-tailed noise into the system model. Based on covariance intersection, a fast CI event-triggered fusion estimator is proposed, and a sufficient condition for error bounded [11] is derived.

However, if the data block is large, this augmentation method imposes a large computational burden. Therefore, the method based on state weighting has gradually attracted the attention of researchers [12]. A high-gain state estimator is proposed for nonlinear asynchronous continuous-discrete sampling systems. The convergence and stability of the estimator are proven by analyzing the Riccati equation [13]. As the research progresses, multiplicative noise and missing measurements have also attracted attention. The estimation problem of asynchronous randomly sampling systems with packet losses is studied, and a centralized fusion algorithm based on measurement data reordering is proposed [14]. Using a neural network compensation model, the optimal fusion estimator based on fault detector is proposed [15]. In recent years, the study of event-triggered mechanisms has become a hot topic due to the need to conserve communication resources [16]. The term “optimal estimation” refers to the best possible estimation of the state or parameter based on a specific estimation criterion when the measurements are known. Based on the minimum entropy error criterion, a sequential fusion estimator is proposed to solve the asynchronous sampling problem with communication delay, and the convergence proof is provided [17]. Unlike traditional estimators designed to handle network-induced phenomena, an estimator that proactively designs communication protocols to mitigate potential non-ideal situations has garnered attention. The estimation problem for asynchronous sampling systems with repetitive processes is considered [18].

Since packet losses are inevitable due to network congestion and connection disruptions, it is necessary to consider packet losses in the estimation problem of asynchronous sampling systems. In practice, nonlinear systems offer more advantages than linear systems in describing the dynamic evolution process of the system. Therefore, the nonlinear asynchronous sampling fusion estimation problem holds greater research value. Compared with the augmentation method, measurement fusion has lower computational complexity, and its accuracy is close to that of the centralized method [19]. Considering the characteristics of asynchronous sampling in network systems, sequential methods are more suitable for dealing with asynchronous fusion problems [20]. In addition, the unmodeled dynamics, such as highly maneuvering systems [21] and anti-modeling [22], are easily overlooked when establishing state models, and the established models are also susceptible to modeling errors. Therefore, an observation inference method based on interpolation is proposed to solve the asynchronous estimation problem. In summary, the main contributions of this paper are as follows:For the state estimation problem of a nonlinear asynchronous sampling system, the synchronization of the state space model is realized by weighting two adjacent state update points at the measurement sampling point.Based on the concept of probability density distribution of an approximate nonlinear function, a deterministic sampling Gaussian estimation framework is proposed, which is considered superior to the algorithm framework with an approximate nonlinear function.Considering the potential modeling errors and unmodeled dynamics in the state equation, a non-recursive estimator based on interpolation operations that solely depends on measurement data is proposed, whose advantage lies in the minimal computational load and rapid calculation speed.

The paper is organized as follows. The system model, the basic theory, and the interested problem are presented in Section 2. The state estimator for an asynchronous sampling system is proposed, and an observation inference scheme independent of the state model is presented in Section 3. The distributed fusion estimator adapted to the multi sensor system is given in Section 4. Section 5 shows the performance analysis of the estimator. Section 6 provides simulation experiments. Section 7 concludes this paper.

**Notation** $R^{n}$ means the n-dimensional Euclidean space. $A^{T}$ and $A^{-1}$ stand for the trace and determinant of a matrix A. P(⋅) and E(⋅) represent the probability and the expectation.

## 2. System Architecture

The asynchronous non-uniform sampling stochastic system with packet dropout is characterized by.
(1)$$x_{k+1}=\Phi x_{k}+B\xi_{k}u_{k}+\Gamma w_{k}$$
(2)$${\bar{y}}_{k_{i}}=\gamma_{k_{i}}\left(h\left(x_{k_{i}}\right)+v_{k_{i}}\right)$$
where $x_{k}\in R^{n}$ denotes the state, $u_{k}\in R^{p}$ means the control input, ${\bar{y}}_{k_{i}}\in R^{m}$ is the measurement, and $x_{k_{i}}$ denotes the state at sampling points ${\bar{y}}_{k_{i}}$. The statistical properties of the noise $w_{k}\in R^{r}$ and $v_{k_{i}}\in R^{m}$ are provided. Φ, B and Γ are matrices with appropriate dimensions.
(3)$$E\left\{\left[\begin{array}{c} w_{k}\\ v_{k_{i}} \end{array}\right]\left[\begin{array}{cc} w_{t}^{T} & v_{t_{i}}^{T} \end{array}\right]\right\}=\left[\begin{array}{cc} Q_{w}\delta_{kt} & 0\\ 0 & Q_{v}\delta_{k_{i}t_{i}} \end{array}\right]$$

The measurement function h(⋅) is the nonlinear smooth function. Random variables following a Bernoulli distribution are used to describe packet dropout. Specifically, $\xi_{k}$ and $\gamma_{k_{i}}$ represent the data losses of control input and measurement end.
(4)$$P\{\xi_{k}=1\}=\;\bar{\xi },\;P\left\{\gamma_{k_{i}}=1\right\}=\;\bar{\gamma }$$

Figure 1 illustrates the general process of information flow in the networked control system. The initial state $x_{0}$ with mean $\mu_{0}$ and variance $P_{0}$ is uncorrelated with the other random variables. Such a linear dynamic nonlinear observation system is common in the field of target tracking [23] and robot localization. The measurement data are usually obtained in the polar coordinate system or spherical coordinate system. For example, the distance and angle of the target are measured in the polar coordinate system, while the state equation is often established in the rectangular coordinate system. The former and the latter are undoubtedly nonlinear relationships [24]. The non-uniform sampling mechanism is illustrated in Figure 2, where the state is updated uniformly, and measurements are randomly sampled.

The difference between non-uniform sampling and uniform sampling but random packet loss lies mainly in two aspects: the sampling strategy and the effects of packet loss. Firstly, in terms of the sampling strategy, non-uniform sampling means that the data points are not collected according to a fixed time interval or frequency, but rather based on some non-uniform or random pattern. This can lead to an uneven distribution of data, with some areas being dense and others sparse. In contrast, uniform sampling but random packet loss involves sampling at a fixed time interval or frequency, but the sampled packets may be randomly lost during transmission or processing. Secondly, regarding the impact of packet loss, although there is random packet loss in both cases, the effect on data quality will vary due to different sampling strategies. In the case of non-uniform sampling, the uneven distribution of data can be exacerbated by packet loss, leading to loss or difficulty in analyzing important information. In the case of uniform sampling, even though the data itself are uniformly distributed, random packet loss can still lead to discontinuity or missing information.

Several measurements are distinguished here for the convenience of deriving subsequent algorithms. The measurement under the non-uniform sampling mechanism is ${\underline{y}}_{k_{i}}=h\left(x_{k_{i}}\right)+v_{k_{i}}$. The non-uniform sampling measurement considering the effect of packet loss is ${\bar{y}}_{k_{i}}=\gamma_{k_{i}}\left(h\left(x_{k_{i}}\right)+v_{k_{i}}\right)$. The successfully received measurement at the estimator is defined as $\left\{y_{k_{i}}\right\}=\left\{{\bar{y}}_{k_{i}},\gamma_{k_{i}}=1\right\}$.

Next, we begin with the measurement equation and propose an estimation algorithm based on pure measurement data by employing interpolation techniques and inversion operations.

## 3. Gaussian Filter at Measurement Sampling Points

The pseudo-state equation is calculated using the state weighting. The recursive estimator at the random sampling point can be derived using the projection theorem.

Let $k_{i}$ and $k_{i-1}$ be the sampling times of the i-th and (i − 1)-th successfully received measurements of the intervals (k−1,k] and (l−1,l], respectively. Define $\alpha_{i}^{k}=k-k_{i}$ and $\alpha_{i-1}^{l}=l-k_{i-1}$, clearly $0\leq \alpha_{i}^{k}<1$ and $0\leq \alpha_{i-1}^{l}<1$.

The state at a time instant $k-\alpha_{i}^{k}$ can be described
(5)$$x_{k-\alpha_{i}^{k}}=\left(1-\alpha_{i}^{k}\right)x_{k}+\alpha_{i}^{k}x_{k-1}$$

This formula describes two scenarios for the position of the sensor output sampling points. In the first case, the sensor output is sampled as soon as the state is updated; therefore, the system state corresponds to the state measured by the sensor. It denotes that $\alpha_{i}^{k}=0$ and $x_{{k-\alpha }_{i}^{k}}=x_{k}$. In the second case, the sensor output is sampled between the state update period, which means that $0\leq \alpha_{i}^{k}<1$ and $x_{{k-\alpha }_{i}^{k}}=\left(1-\alpha_{i}^{k}\right)x_{k}+\alpha_{i}^{k}x_{k-1}$.

Similarly, the state at the sampling time $l-\alpha_{i-1}^{l}$ can be described.
(6)$$x_{l-\alpha_{i-1}^{l}}=\left(1-\alpha_{i-1}^{l}\right)x_{l}+\alpha_{i-1}^{l}x_{l-1}$$

**Lemma 1.** *The state equation at the random sampling point can be obtained through state weighting*(7)$$x_{k-\alpha_{i}^{k}}=\Phi_{i,i-1}^{k,l}x_{l-\alpha_{i-1}^{l}}+U_{i,i-1}^{k,l}\left(k-1,l-1\right)+W_{i,i-1}^{k,l}\left(k-1,l-1\right)$$(8)$$y_{k-\alpha_{i}^{k}}=h\left(x_{k-\alpha_{i}^{k}}\right)+v_{k-\alpha_{i}^{k}}$$*where* $\Phi_{i,i-1}^{k,l}=\beta_{i}^{k}\Phi^{k-l}{\left(\beta_{i-1}^{l}\right)}^{-1}$*,* $\beta_{i}^{k}=\left[\left(1-\alpha_{i}^{k}\right)\Phi +\alpha_{i}^{k}I_{n}\right]$.
(9)$$W_{i,i-1}^{k,l}\left(k-1,l-1\right)=\sum_{j=l}^{k}\Gamma_{i,i-1}^{k,l}\left(j\right)\Gamma w_{j-1}$$(10)$$U_{i,i-1}^{k,l}\left(k-1,l-1\right)=\sum_{j=l}^{k}\Gamma_{i,i-1}^{k,l}\left(j\right)\Gamma \xi_{j-1}u_{j-1}$$(11)$$\Gamma_{i,i-1}^{k,l}\left(j\right)=\{\begin{array}{ll} \beta_{i}^{k}\Phi^{k-1-l}-\Phi_{i,i-1}^{k,l}\left(1-\alpha_{i-1}^{l}\right), & j=l\\ \beta_{i}^{k}\Phi^{k-1-j}, & l<j<k\\ 1-\alpha_{i}^{k}, & j=k\\ \left(1-\alpha_{i}^{k}\right)I_{n}-\Phi_{i,i-1}^{k,l}\left(1-\alpha_{i-1}^{l}\right), & k=l \end{array}$$
*The covariance matrix of the system noise is calculated by*

(12)
$$Q_{W_{i,i-1}^{k,l}}\left(k-1,l-1\right)=\sum_{j=l}^{k}\left(\Gamma_{i,i-1}^{k,l}\left(j\right)\Gamma \right)Q_{w}{\left(\Gamma_{i,i-1}^{k,l}\left(j\right)\Gamma \right)}^{T}$$



**Remark 1.** 
*Unlike existing modeling methods, the approach used in this paper does not increase the dimension of the measurement Equation (12), hence it is referred to as a non-augmented state space model. State weighting refers to the weighted sum of two adjacent state update points to construct the new state, as shown in (5). After synchronization, the state equation can be obtained by iteratively updating the original state equation to maintain consistency with the measurement equation over time 14. However, in reality, there is no state update at the random sampling point, so the state equation is referred to as a pseudo-state equation.*


After obtaining the synchronization model, the state estimators at the sampling points are determined.

**Theorem 1.** 
*For the state space model (7) and (8), the recursive asynchronous estimator is given as follows.*

*The state predictor and its covariance matrix are computed*

(13)
$${\hat{x}}_{k-\alpha_{i}^{k}|l-\alpha_{i-1}^{l}}=\Phi_{i,i-1}^{k,l}{\hat{x}}_{l-\alpha_{i-1}^{l}|l-\alpha_{i-1}^{l}}+U_{i,i-1}^{k,l}\left(k-1,l-1\right)+{\hat{W}}_{i,i-1}^{k,l}\left(k-1,l-1|l-\alpha_{i-1}^{l}\right)$$


(14)
$$\begin{array}{c} P_{k-\alpha_{i}^{k}|l-\alpha_{i-1}^{l}}^{x}=\left(\Phi_{i,i-1}^{k,l}\right)P_{l-\alpha_{i-1}^{l}|l-\alpha_{i-1}^{l}}^{x}{\left(\Phi_{i,i-1}^{k,l}\right)}^{T}+Q_{W_{i,i-1}^{k,l}}\left(k-1,l\right)+\left(\Gamma_{i,i-1}^{k,l}\left(l\right)\Gamma \right)P_{l-1|l-\alpha_{i-1}^{l}}^{w}{\left(\Gamma_{i,i-1}^{k,l}\left(l\right)\Gamma \right)}^{T}\\ +\Gamma_{i,i-1}^{k,l}\left(l\right)\Gamma P_{l-1,l-\alpha_{i-1}^{l}|l-\alpha_{i-1}^{l}}^{wx}{\left(\Phi_{i,i-1}^{k,l}\right)}^{T}+{\left\{\Gamma_{i,i-1}^{k,l}\left(l\right)\Gamma P_{l-1,l-\alpha_{i-1}^{l}|l-\alpha_{i-1}^{l}}^{wx}{\left(\Phi_{i,i-1}^{k,l}\right)}^{T}\right\}}^{T} \end{array}$$


*The state and noise gain matrices are calculated*

(15)
$$K_{k-\alpha_{i}^{k}}^{x}=P_{k-\alpha_{i}^{k}|l-\alpha_{i-1}^{l}}^{xy}{\left(P_{k-\alpha_{i}^{k}}^{y}\right)}^{-1}$$


(16)
$$K_{k-1|k-\alpha_{i}^{k}}^{w}=P_{k-1,k-\alpha_{i}^{k}|l-\alpha_{i-1}^{l}}^{wy}{\left(P_{k-\alpha_{i}^{k}}^{y}\right)}^{-1}$$


*State and noise estimators are calculated.*

(17)
$${\hat{x}}_{k-\alpha_{i}^{k}|k-\alpha_{i}^{k}}={\hat{x}}_{k-\alpha_{i}^{k}|l-\alpha_{i-1}^{l}}+K_{k-\alpha_{i}^{k}}^{x}{\tilde{y}}_{k-\alpha_{i}^{k}|l-\alpha_{i-1}^{l}}$$


(18)
$${\hat{w}}_{k-\alpha_{i}^{k}|k-\alpha_{i}^{k}}={\hat{w}}_{k-1|l-\alpha_{i-1}^{l}}+K_{k-1|k-\alpha_{i}^{k}}^{w}{\tilde{y}}_{k-\alpha_{i}^{k}|l-\alpha_{i-1}^{l}}$$

*where innovation is computed.*

(19)
$${\tilde{y}}_{k-\alpha_{i}^{k}|l-\alpha_{i-1}^{l}}=y_{k-\alpha_{i}^{k}}-\int h\left(x_{k-\alpha_{i}^{k}}\right)N\left(x_{k-\alpha_{i}^{k}};x_{k-\alpha_{i}^{k}|l-\alpha_{i-1}^{l}},P_{k-\alpha_{i}^{k}|l-\alpha_{i-1}^{l}}^{x}\right)dx_{k-\alpha_{i}^{k}}$$


*The covariance matrices for state and noise are given.*

(20)
$$P_{k-\alpha_{i}^{k}|k-\alpha_{i}^{k}}^{x}=P_{k-\alpha_{i}^{k}|l-\alpha_{i-1}^{l}}^{x}-K_{k-\alpha_{i}^{k}}^{x}P_{k-\alpha_{i}^{k}}^{y}{\left(K_{k-\alpha_{i}^{k}}^{x}\right)}^{T}$$


(21)
$$P_{k-1|k-\alpha_{i}^{k}}^{w}=P_{k-1|l-\alpha_{i-1}^{l}}^{w}-K_{k-1|k-\alpha_{i}^{k}}^{w}P_{k-\alpha_{i}^{k}}^{y}{\left(K_{k-1|k-\alpha_{i}^{k}}^{w}\right)}^{T}$$



**Proof of Theorem 1.** First, we need to prove the time updating part. The prediction error equation can be obtained by
(22)$${\tilde{x}}_{k-\alpha_{i}^{k}|l-\alpha_{i-1}^{l}}=\Phi_{i,i-1}^{k,l}{\tilde{x}}_{l-\alpha_{i-1}^{l}|l-\alpha_{i-1}^{l}}+W_{i,i-1}^{k,l}\left(k-1,l\right)+\Gamma_{i,i-1}^{k,l}\left(l\right)\Gamma {\tilde{w}}_{l-1|l-\alpha_{i-1}^{l}}$$

Substituting (22) into $P_{k-\alpha_{i}^{k}|l-\alpha_{i-1}^{l}}^{x}=E\left\{{\tilde{x}}_{k-\alpha_{i}^{k}|l-\alpha_{i-1}^{l}}{\tilde{x}}_{k-\alpha_{i}^{k}|l-\alpha_{i-1}^{l}}^{T}\right\}$ leads to (14).

Calculating the gain matrix $K_{k-\alpha_{i}^{k}}^{x}$ by definition (15) requires first calculating the covariance matrices $P_{k-\alpha_{i}^{k}|l-\alpha_{i-1}^{l}}^{xy}$ and $P_{k-\alpha_{i}^{k}}^{y}$.
(23)$$P_{k-\alpha_{i}^{k}|l-\alpha_{i-1}^{l}}^{xy}=\int x_{k-\alpha_{i}^{k}}h^{T}\left(x_{k-\alpha_{i}^{k}}\right)N_{x}dx_{k-\alpha_{i}^{k}}-{\hat{x}}_{k-\alpha_{i}^{k}|l-\alpha_{i-1}^{l}}{\hat{y}}_{k-\alpha_{i}^{k}|l-\alpha_{i-1}^{l}}^{T}$$(24)$$P_{k-\alpha_{i}^{k}}^{y}=\int h\left(x_{k-\alpha_{i}^{k}}\right)h^{T}\left(x_{k-\alpha_{i}^{k}}\right)N_{x}dx_{k-\alpha_{i}^{k}}+Q_{v}-{\hat{y}}_{k-\alpha_{i}^{k}|l-\alpha_{i-1}^{l}}{\hat{y}}_{k-\alpha_{i}^{k}|l-\alpha_{i-1}^{l}}^{T}$$(25)$$N_{x}=N(x_{k-\alpha_{i}^{k}};x_{k-\alpha_{i}^{k}|l-\alpha_{i-1}^{l}},P_{k-\alpha_{i}^{k}|l-\alpha_{i-1}^{l}}^{x})$$

Similarly, since the innovation covariance matrix $P_{k-\alpha_{i}^{k}}^{y}$ is computed, computing the smoother gain matrix $K_{k-1|k-\alpha_{i}^{k}}^{w}$ requires computing only the covariance matrix
(26)$$P_{k-1,k-\alpha_{i}^{k}|l-\alpha_{i-1}^{l}}^{wy}=\int w_{k-1}h^{T}\left(x_{k-\alpha_{i}^{k}}\right)N_{wx}d\left[\begin{array}{c} w_{k-1}\\ x_{k-\alpha_{i}^{k}} \end{array}\right]-{\hat{w}}_{k-1|l-\alpha_{i-1}^{l}}{\hat{y}}_{k-\alpha_{i}^{k}|l-\alpha_{i-1}^{l}}^{T}$$(27)$$N_{wx}=N(\left[\begin{array}{c} w_{k-1}\\ x_{k-\alpha_{i}^{k}} \end{array}\right];\left[\begin{array}{c} {\hat{w}}_{k-1|l-\alpha_{i-1}^{l}}\\ {\hat{x}}_{k-\alpha_{i}^{k}|l-\alpha_{i-1}^{l}} \end{array}\right],\left[\begin{array}{cc} P_{k-1|l-\alpha_{i-1}^{l}}^{w} & P_{k-1,k-\alpha_{i}^{k}|l-\alpha_{i-1}^{l}}^{wx}\\ {\left(P_{k-1,k-\alpha_{i}^{k}|l-\alpha_{i-1}^{l}}^{wx}\right)}^{T} & P_{k-\alpha_{i}^{k}|l-\alpha_{i-1}^{l}}^{x} \end{array}\right])$$
where the cross-covariance matrix $P_{k-1,k-\alpha_{i}^{k}|l-\alpha_{i-1}^{l}}^{wx}$ for noise and state is computed by
(28)$$P_{k-1,k-\alpha_{i}^{k}|l-\alpha_{i-1}^{l}}^{wx}=\{\begin{array}{cc} Q_{w}\Gamma^{T}{\left(\Gamma_{i,i-1}^{k,l}\left(k\right)\right)}^{T}, & k>l\\ P_{l-1,l-\alpha_{i-1}^{l}|l-\alpha_{i-1}^{l}}^{wx}{\left(\Phi_{i,i-1}^{k,l}\right)}^{T}+P_{l-1|l-\alpha_{i-1}^{l}}^{w}\Gamma^{T}{\left(\Gamma_{i,i-1}^{k,l}\left(l\right)\right)}^{T}, & k=l \end{array}$$

Next, the proof of the measurement update part is presented. Based on the definition of the filtering error equation, ${\tilde{x}}_{k-\alpha_{i}^{k}|k-\alpha_{i}^{k}}$ can be concluded
(29)$${\tilde{x}}_{k-\alpha_{i}^{k}|k-\alpha_{i}^{k}}={\tilde{x}}_{k-\alpha_{i}^{k}|l-\alpha_{i-1}^{l}}-K_{k-\alpha_{i}^{k}}^{x}{\tilde{y}}_{k-\alpha_{i}^{k}|l-\alpha_{i-1}^{l}}$$

The prediction error covariance matrix (20) can be obtained by $P_{k-\alpha_{i}^{k}|k-\alpha_{i}^{k}}^{x}=E\left\{{\tilde{x}}_{k-\alpha_{i}^{k}|k-\alpha_{i}^{k}}{\tilde{x}}_{k-\alpha_{i}^{k}|k-\alpha_{i}^{k}}^{T}\right\}$. The projection theorem demonstrates that the noise smoother possesses a structure similar to that of the state estimator, allowing the derivation of the smoothing error equation.
(30)$${\tilde{w}}_{k-1|k-\alpha_{i}^{k}}={\tilde{w}}_{k-1|l-\alpha_{i-1}^{l}}-K_{k-1|k-\alpha_{i}^{k}}^{w}{\tilde{y}}_{k-\alpha_{i}^{k}|l-\alpha_{i-1}^{l}}$$

Similarly, (21) and (33) can be obtained through the cross-covariance matrix $P_{k-1|k-\alpha_{i}^{k}}^{w}=E\left\{{\tilde{w}}_{k-1|k-\alpha_{i}^{k}}{\tilde{w}}_{k-1|k-\alpha_{i}^{k}}^{T}\right\}$ and $P_{k-1,k-\alpha_{i}^{k}|k-\alpha_{i}^{k}}^{wx}=E\left\{{\tilde{w}}_{k-1|k-\alpha_{i}^{k}}{\tilde{x}}_{k-\alpha_{i}^{k}|k-\alpha_{i}^{k}}^{T}\right\}$.
(31)$$P_{k-1,k-\alpha_{i}^{k}|k-\alpha_{i}^{k}}^{wx}=P_{k-1,k-\alpha_{i}^{k}|l-\alpha_{i-1}^{l}}^{wx}-K_{k-1|k-\alpha_{i}^{k}}^{w}P_{k-\alpha_{i}^{k}}^{y}{\left(K_{k-\alpha_{i}^{k}}^{x}\right)}^{T}$$

The corresponding nonlinear estimator can be obtained through various deterministic sampling methods. For instance, the Unscented Kalman Filter (UKF) can be derived through unscented transformation, while the Cubature Kalman Filter (CKF) can be obtained by using the spherical radial cubature rule. The non-augmented model-based UKF and CKF algorithms proposed in this section are referred to as NM-UKF and NM-CKF algorithms. □

**Remark 2.** *If the sensor output is sampled as soon as the state is updated, the filter* ${\hat{x}}_{k-\alpha_{i}^{k}|k-\alpha_{i}^{k}}$ *becomes the estimator* ${\hat{x}}_{k|k}$ *at time* k*. If the measurement data are not sampled at time* k *but within the period* (k−1,k]*, the predictor* ${\hat{x}}_{k|k-\alpha_{i}^{k}}$ *based on the filter* ${\hat{x}}_{k-\alpha_{i}^{k}|k-\alpha_{i}^{k}}$ *is used as the estimator* ${\hat{x}}_{k|k}$ *at time* k*. If the sensor output is not sampled in time after state update, the predictor* ${\hat{x}}_{k|k-1}$ *is used as the estimator* ${\hat{x}}_{k|k}$ *at time* k.

**Remark 3.** 
*However, in reality, obtaining an accurate state model is challenging due to the presence of unmodeled dynamics. This, in turn, leads to a performance deterioration of conventional filters that rely on the accuracy of the model. Therefore, a measurement inference scheme based on interpolation is also proposed.*


First, measurement data at the state update point can be obtained using the piecewise cubic Hermite interpolation algorithm. (It is worth noting that, compared with cubic spline interpolation [25], the modified Akima piecewise cubic interpolation (MAPCI) produces fewer fluctuations and is more suitable for handling rapid changes between platform regions. Compared to the shape-preserving piecewise cubic interpolation [26], MAPCI is less sharply flattened, which enables it to effectively handle oscillating data.) The next step is to estimate the state using measurements that are synchronized with the state updates. In other words, the state is inferred directly from the measurements, which are corrupted by noise [27]. (The first scenario involves an invertible observation function, but the observation noise is unknown. In this case, the observation noise may be colored noise, correlated noise, non-Gaussian noise, or other complex situations. Since it cannot be accurately described, we choose to ignore the noise to simplify the problem. The second scenario occurs when the observation function is invertible, and the observation noise is known [28]. If the system model adheres to the assumptions of the classical Kalman filter, then the inversion operation can be implemented directly. If the observation function is nonlinear and the observation noise is complex [29], it may be more reasonable to sample the observation noise and use the mean and variance of a set of samples to characterize the state estimate. The third scenario occurs when the observation function is not invertible. Facing this situation in practical engineering, a common solution is to enhance observability by increasing the number of sensors [30]. The addition of sensors to address the irreversibility of the observation function mainly depends on the number of observation variables). A summary of the pure measurement methods used to solve the multirate estimation problem can be found in Appendix A.

The block diagram of our proposed Gaussian estimator with the O_2_ inference based on Gaussian Hermite interpolation can be seen in Figure 3. The concept behind both methods is to convert the asynchronous sampling system into a synchronous sampling system and then use an appropriate synchronous system scheme to derive the state estimation. The synchronization concept of the Gaussian estimator relies on the timing of measurement sampling to achieve synchronization through state weighting. Subsequently, a Gaussian estimation framework is introduced based on the concept of a probability density function that approximates the nonlinear function. The synchronization concept of O_2_ inference based on Gaussian Hermite interpolation involves aligning the state update time through Gaussian Hermite interpolation, followed by deriving the state estimate using O_2_ inference.

## 4. Distributed Fusion Estimator

In the previous section, we proposed an asynchronous sampling state estimation algorithm. In this section, we provide an asynchronous fusion estimator that is suitable for multi-source systems.

For multi-sensor systems, the state equation is (1). The measurement equation of the multi-sensor random sampling system can be given as follows.
(32)$$y_{k_{i}^{l}}^{l}=\gamma_{k_{i}^{l}}^{l}\left(h^{l}\left(x_{k_{i}^{l}}^{l}\right)+v_{k_{i}^{l}}^{l}\right),l=1,\cdots ,L$$
where $y_{k_{i}^{l}}^{l}$ represents the ith measurement of sensor j, and $v_{k_{i}^{l}}^{l}$ denotes the noise, which is uncorrelated with the other random variables. $\gamma_{k_{i}^{l}}^{l}$ is a random variable that follows the Bernoulli distribution with mean $E\left\{\gamma^{l}\right\}={\bar{\gamma }}^{l}$, ${\bar{\gamma }}^{l}\in \left[0,1\right]$.

When considering the multi-sensor system, an intuitive approach is to rearrange all measurements. This approach has the advantage of treating the system to some extent as a single sensor system. In this case, the proposed estimation algorithm in this paper can be used to solve the problem of asynchronous fusion estimation. However, the drawback of this scheme is that it requires all sensors to function properly, which means that the reliability of this scheme is not optimal. In other words, if any sensor fails, the estimator diverges. To solve the issue of asynchronous sampling in fusion estimation, we propose distributed fusion estimation algorithms.

To avoid calculating the cross-covariance matrix [31], the CI fusion algorithm is adopted, which gives an upper bound on the error variance matrix by solving the optimization problem online. The state estimator obtained by the CI fusion algorithm can be expressed as
(33)$${\hat{x}}_{k|k}^{\mathrm{CI}}=\sum_{l=1}^{L}\omega_{k}^{l}P_{k|k}^{\mathrm{CI}}{\left(P_{k|k}^{l}\right)}^{-1}{\hat{x}}_{k|k}^{l}$$
(34)$$P_{k|k}^{\mathrm{CI}}={\left[\sum_{l=1}^{L}\omega_{k}^{l}{\left(P_{k|k}^{l}\right)}^{-1}\right]}^{-1}$$
where $P_{k|k}^{\mathrm{CI}}$ is the error covariance matrix of the fusion estimator. $\omega_{k}^{l}\left(0\leq \omega_{k}^{l}\leq 1\right)$ is the weighting coefficient, which is obtained by solving the following optimization problem
(35)$$\underset{\sum_{l=1}^{L}\omega_{k}^{l}=1}{\min}J=\underset{\sum_{l=1}^{L}\omega_{k}^{l}=1}{\min}\mathrm{tr}\left(P_{k|k}^{\mathrm{CI}}\right)$$

The distributed fusion estimator presented in this paper can be adapted to non-ideal situations because some of its operations can be performed within the sensor. Meanwhile, it has good fault tolerance, flexibility, and reliability due to its low requirement for channel capacity.

## 5. Performance Analysis

We first analyze the estimation performance when there are no packet losses.

**Lemma 2.** *Assuming that the estimator* ${\hat{x}}_{k-1|k-1}$ *and variance matrix* $P_{k-1|k-1}$ *of state* $x_{k-1}$ *have been obtained,* ${\hat{x}}_{k|k}$ *and* ${\underline{\hat{x}}}_{k|k}$ *represent the estimator of state* $x_{k}$ *at time* k *obtained by using* $N_{k}$ *or* ${\underline{N}}_{k}$ *data in the period* (k−1,k] *, respectively. The corresponding variance* $P_{k|k}$ *and* ${\underline{P}}_{k|k}$*. So, if* ${\underline{N}}_{k}\geq N_{k}\geq 0$ *, then* ${\underline{P}}_{k|k}\leq P_{k|k}$.

Lemma 2 states that the more fully the measurement data are used over time, the more accurate the estimation result. It also shows that when the local estimation error is bounded, the fused estimation error is also bounded. Therefore, in the following, we analyze the performance of the local estimator

**Definition 1** (Stochastic Boundedness [32])**.** $P_{k|k-1}^{\gamma }$ *is stochastically bounded if*
M *exists such that*
(36)$$\underset{M\to \infty }{\lim}\sup P\left\{\left\|P_{k|k-1}^{\gamma }\right\|>M\right\}=0$$

**Definition 2** (Mean Boundedness [32])**.** $P_{k|k-1}^{\gamma }$ *is mean bounded if* $M^{\bar{\gamma }{,P}_{0}}$ *exists such that*
(37)$$\sup E\left\{P_{k|k-1}^{\gamma }\right\}\leq M^{\bar{\gamma }{,P}_{0}}$$

**Lemma 3.** *For a matrix pair* $\left(\Phi ,\sqrt{Q}\right)$ *is stabilizable.*(1)$P_{k|k-1}^{\xi }$ *is mean boundedness for* $\forall P_{0}>0$ *and* $1-\frac{1}{\lambda^{2}}<\bar{\xi }\leq 1$*, where* λ *is the maximum eigenvalue of* Φ.(2)$P_{k|k-1}^{\xi }$ *is stochastic boundedness for* $\forall P_{0}>0$ *and* $0<\bar{\xi }\leq 1$.

**Theorem 2.** *For matrix pair* $\left(\Phi ,\sqrt{Q}\right)$ *to be stabilizable, define*  $P\{\gamma_{k}\neq 1|\xi_{k}=1\}=\alpha$ *and*  $P\{\gamma_{k}=1|\xi_{k}\neq 1\}=\beta$*, if* α+β<1*, then*
(1)$P_{k|k-1}^{\gamma }$ *is mean boundedness for* $\forall P_{0}>0$ *and* $\beta \leq \bar{\gamma }<\beta +\frac{\lambda^{2}}{1-\alpha -\beta }$.(2)$P_{k|k-1}^{\gamma }$ *is stochastic boundedness for* $\forall P_{0}>0$ *and* $\beta \leq \bar{\gamma }<1-\alpha$.

**Proof of Theorem 2.** It follows from the Bayesian formula
(38)$$\begin{array}{ll} \bar{\gamma } & =P\left\{\gamma_{k}=1\right\}\\ & =P\left\{\gamma_{k}=1|\xi_{k}=1\right\}P\left\{\xi_{k}=1\right\}+P\left\{\gamma_{k}=1|\xi_{k}\neq 1\right\}P\left\{\xi_{k}\neq 1\right\}\\ & =\bar{\xi }\left(1-\alpha \right)+\beta \left(1-\bar{\xi }\right)\\ & =\bar{\xi }\left(1-\alpha -\beta \right)+\beta \end{array}$$
The error covariance matrix is given by
(39)$$\begin{array}{ll} P_{k+1|k}^{\gamma } & =\Phi P_{k|k-1}^{\gamma }\Phi^{T}+Q-\gamma_{k}\Phi P_{k|k-1}^{xy}{\left(P_{k|k-1}^{yy}\right)}^{-1}{\left(P_{k|k-1}^{xy}\right)}^{T}\Phi^{T}\\ & =\Phi P_{k|k-1}^{\gamma }\Phi^{T}+Q-\xi_{k}\Phi P_{k|k-1}^{xy}{\left(P_{k|k-1}^{yy}\right)}^{-1}{\left(P_{k|k-1}^{xy}\right)}^{T}\Phi^{T}\\ & +\left(\xi_{k}-\gamma_{k}\right)\Phi P_{k|k-1}^{xy}{\left(P_{k|k-1}^{yy}\right)}^{-1}{\left(P_{k|k-1}^{xy}\right)}^{T}\Phi^{T}\\ & \leq \Phi P_{k|k-1}^{\gamma }\Phi^{T}+Q+\left(1-\xi_{k}\right)\Phi P_{k|k-1}^{xy}{\left(P_{k|k-1}^{yy}\right)}^{-1}{\left(P_{k|k-1}^{xy}\right)}^{T}\Phi^{T} \end{array}$$
where
$$P_{k|k-1}^{xy}=\int x_{k}h_{k}^{T}\left(x_{k}\right)N\left(x_{k};{\hat{x}}_{k|k-1},P_{k|k-1}\right)dx_{k}-{\hat{x}}_{k|k-1}{\hat{y}}_{k|k-1}^{T}$$
$$P_{k|k-1}^{yy}=\int h_{k}\left(x_{k}\right)h_{k}^{T}\left(x_{k}\right)N\left(x_{k};{\hat{x}}_{k|k-1},P_{k|k-1}\right)dx_{k}-{\hat{y}}_{k|k-1}{\hat{y}}_{k|k-1}^{T}+R$$It follows from Lemma 3 that $P_{k|k-1}^{\gamma }$ is mean boundedness if
(40)$$1-\frac{1}{\lambda^{2}}<1-\bar{\xi }\leq 1$$$P_{k|k-1}^{\gamma }$ is stochastic boundedness if
(41)$$0<1-\bar{\xi }\leq 1$$It follows from (38), that we have
(42)$$1-\bar{\xi }=\frac{1-\alpha -\bar{\xi }}{1-\alpha -\beta }$$Substituting (42) into (40) and (41) yields Theorem 2. □

## 6. Simulations

### 6.1. Target Tracking System

State estimation algorithms are widely used in target tracking, industrial control, wireless communication, and other fields. Among these systems, the target tracking system is often used to validate the effectiveness of estimation algorithms. The motion model is a crucial component of a target tracking algorithm. Most current target tracking algorithms rely on the motion model. In the research into target tracking algorithms, several simple and easily implementable motion models have been the focus of development work. The constant acceleration model is one of the most common basic motion models. The range-bearing model of active radar is one of the most common measurement models, where measurements include the range and bearing, which are affected by noise. Therefore, this paper uses the following discrete-time asynchronous random sampling system as the verification example in the simulation section.
(43)$$\Phi =\left[\begin{array}{cccc} 1 & \Delta & 0 & 0\\ 0 & 1 & 0 & 0\\ 0 & 0 & 1 & \Delta \\ 0 & 0 & 0 & 1 \end{array}\right],\;B=\left[\begin{array}{ll} \Delta^{2}/2 & 0\\ \Delta & 0\\ 0 & \Delta^{2}/2\\ 0 & \Delta \end{array}\right],\;\Gamma =\left[\begin{array}{ll} \Delta^{3}/6 & 0\\ \Delta^{2}/2 & 0\\ 0 & \Delta^{3}/6\\ 0 & \Delta^{2}/2 \end{array}\right]$$
where $x_{k}={\left[p_{x_{k}},{\dot{p}}_{x_{k}},p_{y_{k}},{\dot{p}}_{y_{k}}\right]}^{T}$, ${\left[p_{x_{k}},{\dot{p}}_{x_{k}}\right]}^{T}$ and ${\left[p_{y_{k}},{\dot{p}}_{y_{k}}\right]}^{T}$ is the position and velocity, respectively. $u_{k}={\left[u_{1,k},u_{2,k}\right]}^{T}$ and $w_{k}={\left[w_{1,k},w_{2,k}\right]}^{T}$ denote the control input and system noise. The variances of the random variables $w_{1,k}$ and $w_{2,k}$ are $Q_{w_{1}}=0.09$ and $Q_{w_{2}}=0.25$. The state update period Δ=1s. Let N be the sensor output sampling times in the state update cycle.

The range-bearing measurement equation is given by.
(44)$$y_{k_{i}^{l}}^{l}=\gamma_{k_{i}^{l}}^{l}\left[\begin{array}{c} r_{k_{i}^{l}}^{l}\\ \theta_{k_{i}^{l}}^{l} \end{array}\right]=\gamma_{k_{i}^{l}}^{l}\left(\left[\begin{array}{c} \sqrt{{\left(p_{x_{k_{i}^{l}}}^{l}-S_{x}\right)}^{2}+{\left(p_{y_{k_{i}^{l}}}^{l}-S_{y}\right)}^{2}}\\ \arctan\left(\frac{p_{y_{k_{i}^{l}}}^{l}-S_{y}}{p_{x_{k_{i}^{l}}}^{l}-S_{x}}\right) \end{array}\right]+v_{k_{i}^{l}}^{l}\right)$$
where the active sensor is located at $\left[S_{x},S_{y}\right]$. Set $\bar{\xi }=0.8$, ${\bar{\gamma }}^{1}=0.8$, ${\bar{\gamma }}^{2}=0.9$. Similar to system noise, the variances in measurement noise $v_{k_{i}}^{1}$ and $v_{k_{i}}^{2}$ are $\mathrm{diag}\left(\left[Q_{r^{1}},Q_{\theta^{1}}\right]\right)$ and $\mathrm{diag}\left(\left[Q_{r^{2}},Q_{\theta^{2}}\right]\right)$, and among them $Q_{r^{1}}=1$, $Q_{\theta^{1}}=\pi /90$, $Q_{r^{2}}=0.5$ and $Q_{\theta^{2}}=\pi /100$. The initial values are $x_{0}={\left[0,3,0,-3\right]}^{T}$ and $P_{0}=0.1I_{4}$.

Figure 4 illustrates the position tracking effect of the estimators for sensor 1. The accumulative mean square error (AMSE) of the estimation algorithms for sensor 1 is plotted in Figure 5 and the performance for different estimators is given in Table 1. The core of the Extended Kalman Filter (EKF) lies in the first-order linearization of the nonlinear function. Therefore, for plants with strong nonlinearity, the filtering effect is not ideal, or even unusable. Both the UKF and CKF adhere to the idea that “approximating the probability density function distribution of a nonlinear function is easier than approximating the nonlinear function itself,” and they belong to the Gaussian filtering framework. The UKF algorithm obtains the sigma points and corresponding weights through unscented transformation. The weights are generally negative in high-dimensional systems, which introduces high-order truncation error terms and reduces the accuracy of the algorithm [33]. The CKF algorithm acquires cubature points and propagates them using nonlinear equations. As a result, the weights are always positive, leading to a significant reduction in error. Therefore, in the simulation results, the CKF algorithm demonstrates higher numerical stability and filtering accuracy compared to the UKF algorithm.

However, obtaining the exact state equation is very challenging due to unmodeled dynamics in practical engineering. Therefore, this paper proposes a measurement inference method that does not rely on the state equation. First, timestamp registration is achieved through interpolation. Second, the system state is inferred directly through the O_2_ inference. This process avoids the estimation error caused by unmodeled dynamics in the state equation. According to the simulation results, the MAPCI-O_2_ algorithm and NM-CKF algorithm demonstrate equivalent tracking performance. Moreover, since the MAPCI-O_2_ algorithm relies solely on measurement data, the running time and computational burden of the algorithm are significantly reduced compared to traditional filtering algorithms.

Figure 6 illustrates the feasibility of the proposed estimation algorithm. The AMSEs of local and fusion estimation algorithms are shown in Figure 7 and a performance comparison for local and fusion estimators is given in Table 2. The accuracy of the CI fusion estimation algorithm based on multi-sensor measurement data is higher than that based on single-sensor data.

Figure 8 shows the error curve of the estimator with different N for sensor 1, where N=1/N means that the sensor output is sampled once in the N state update cycle. Simulation results indicate that increasing the number of sampling times during the same state update period leads to a more accurate estimation effect.

The recursive estimation algorithm proposed in Section IV is a general framework, which means that most nonlinear filtering algorithms based on deterministic sampling can be applied within this framework. The UKF and CKF are typical representatives of deterministic sampling used in simulation experiments. Furthermore, the estimation error is used as the performance indicator. Simulation examples demonstrate that the NM-CKF algorithm exhibits the best performance, while the NM-UKF algorithm is slightly inferior, but both algorithms are capable of tracking the system state. However, the NM-EKF algorithm is unable to track the system state. This highlights the necessity and effectiveness of designing the proposed algorithm. When the state equation contains significant uncertainty or is difficult to obtain, the MAPCI-O_2_ algorithm can be used as an effective complement to the traditional state estimator. In addition, in Section V, we extend the state estimation algorithm to include asynchronous sampling with packet losses in the multi-sensor system. Simulation results further confirm the effectiveness of the fusion algorithm.

### 6.2. Spring Mass System

The spring mass system, which is used as a practical example in many studies, is modeled as
$$x_{k+1}=\left[\begin{array}{cc} 0 & 1\\ -\frac{\kappa }{M} & -\frac{c}{M} \end{array}\right]x_{k}+\left[\begin{array}{c} 0\\ \frac{1}{M} \end{array}\right]u_{k}+w_{k}$$
$$y_{k}=\alpha x_{1,k}^{2}+v_{k}$$
where $x_{k}={\left[x_{1,k},x_{2,k}\right]}^{T}$, $x_{1,k}$ and $x_{2,k}$ are the position and velocity of mass. κ is the spring constant. c is the damping coefficient. α is the sensor scale factor.

The system parameters are set as follows: M=1, κ=1, c=0.5 and α=1. The packet losses phenomenon, which is considered a common network induced phenomenon in networked control systems, is described as the Bernoulli distribution. Set $\bar{\xi }=0.8$, and $Q_{w}=\mathrm{diag}\left(\left[1,1\right]\right)$. The initial state is $x_{0}={\left[0,0\right]}^{T}$. Set $\bar{\gamma }=0.8$, and $Q_{v}=4$.

Figure 9 and Figure 10 illustrate the tracking results of different estimators, while Table 3 displays the performance of different estimators. Similar to the conclusion for the target tracking system, the three estimators can approximately track the target. Specifically, the performance of UKF and CKF is similar, while the MAPCI-O_2_ algorithm is slightly outperforms the Gaussian estimator, which may be attributed to the fact that the disturbance induced by measurement noise is partially mitigated by random sampling in the simulation.

To illustrate the impact of the packet loss phenomenon on the estimator, it is necessary to conduct a packet loss rate experiment, and it should be noted that this paper considers both the packet loss $\xi_{k}$ from the controller to the actuator and the packet loss $\gamma_{k}$ from the sensor to the estimator (SE). However, if $\xi_{k}$ changes, it will cause the system model to change, which means that the performance comparison of the estimator loses the benchmark at that time. To this end, we mainly focus on the packet loss phenomenon on the SE side and use NM-CKF as the base algorithm, Figure 11 shows the estimation performance of the estimator under different packet loss rates.

It can be seen that the larger $\bar{\gamma }$ is, the smaller packet losses probability is, and the estimation performance of the filter is better. This means that the phenomenon of packet losses does affect the estimation performance of the filter. The primary focus of this paper is to solve the state estimation problem in asynchronous non-uniform sampling systems. The paper does not employ a compensation strategy to rectify the accuracy losses resulting from packet losses. The existing packet loss compensation strategies mainly include zero compensation, hold compensation, and prediction compensation [34]. Among these, the prediction compensation strategy has the best effect. The main reason is that the predicted value of the measurement based on the state prediction is closer to the current time measurement than the zero and the last time measurement [35]. In addition to filters, there are also observers that can help solve the state estimation problem, among which the intermediate observer is particularly advantageous for resolving packet loss issues [36]. This also motivates us to focus on the fusion estimation problem for asynchronous non-uniform sampling with data packet loss in our future work. Specifically, we aim to investigate more effective and general packet loss compensation strategies and synchronization methods.

In order to verify the applicability of the covariance crossover idea, the distributed fusion estimator experiment is carried out next, where the simulation parameters are ${\bar{\gamma }}^{1}=0.9$ for sensor 1 and ${\bar{\gamma }}^{2}=0.8$ for sensor 2, and the rest of the parameters are the same as above

Figure 12 displays a comparison diagram of estimations between the local estimator and the fusion estimator. It can be seen that the accuracy of the fusion estimator is better than that of the local estimator, indicating that the proposed algorithm remains effective in the context of multi-sensor systems. From the local estimator, it can be seen that a smaller packet loss rate leads to less information loss and higher accuracy of the estimator, as also depicted in Figure 13.

In general, multi-sensor fusion can usually be divided into three architectures: centralized [37], decentralized [38], and distributed [39]. In centralized fusion, the observation equations of all sensors are combined into a high-dimensional observation equation using the dimension expansion method, and then Kalman filtering is applied simultaneously with the state equation. Although centralized fusion can theoretically achieve the global optimal fusion estimate, it comes with the drawbacks of high computational burden and poor fault tolerance [40]. Decentralized fusion uses a weighted local Kalman filter to obtain the fusion estimate, so it is also referred to as weighted fusion. It has the advantages of low computational burden, easy fault diagnosis, and separation. However, such architectures typically necessitate highly reliable sensors, which are often costly and not easily scalable. In addition, due to physical constraints, such as communication delays, the communication bandwidth is limited, and the fusion center cannot effectively communicate with all sensors in a large-scale sensor network [41]. Each sensor node in the distributed architecture uses only the information obtained from its locally connected neighbors for fusion. This feature can offer improved built-in redundancy compared to the other two types of architecture, thereby enhancing robustness against sensor failures. The distributed fusion architecture can also reduce the communication burden because the function of the fusion center is amortized over the nodes [42]. Therefore, this can improve flexibility. The main indicators used in the algorithm evaluation include global optimality, local consistency, fully distributed and communication burden [43], and the selection of different performance indicators results in different advantages and disadvantages of the algorithm, which also encourages us to conduct more comprehensive and detailed research on the fusion architecture in the future.

## 7. Conclusions

We investigate the asynchronous multirate estimator with packet dropout. The state equation at the sampling point is determined based on state weighting. The estimators at the sampling point of the measurement and the state update point are determined using the orthogonal projection theorem and innovation analysis technique. In addition, if the state equation cannot be obtained, a state estimation algorithm based on measurement inference is proposed. The fusion form of the proposed algorithm suitable for a multi-sensor system is provided, and the feasibility of the algorithm is confirmed with simulation experiments.

## Figures and Tables

**Figure 1 sensors-24-02882-f001:**
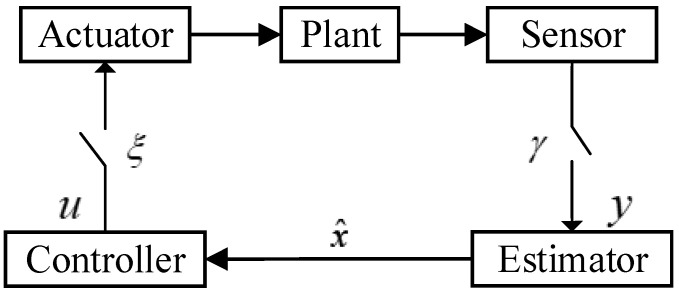
Networked control system under transmission control protocol.

**Figure 2 sensors-24-02882-f002:**
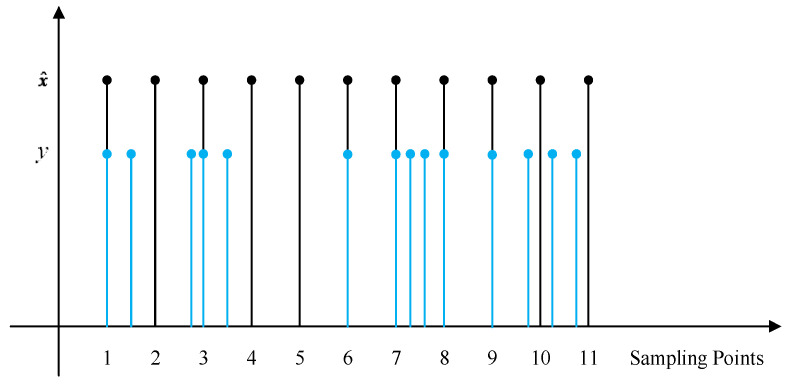
Illustration of non-uniform sampling.

**Figure 3 sensors-24-02882-f003:**
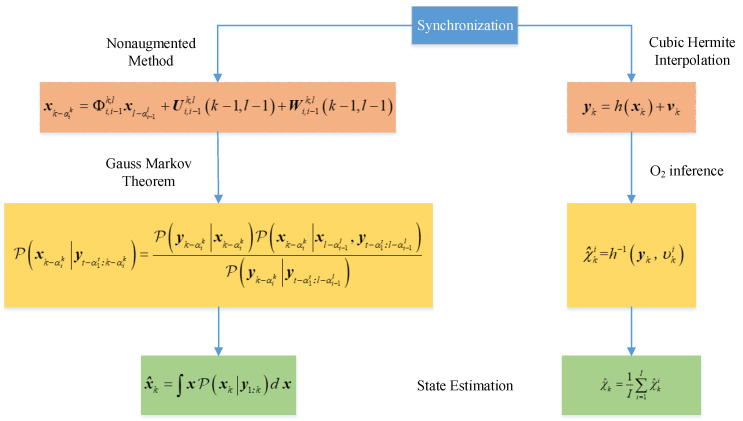
Comparison of Gaussian estimator based on measurement sampling time and O_2_ inference based on state update time.

**Figure 4 sensors-24-02882-f004:**
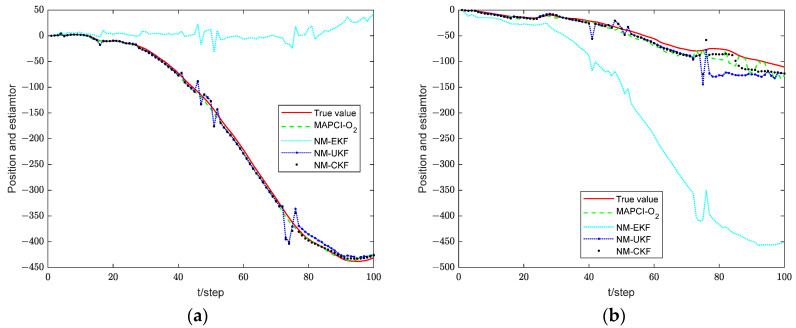
Tracking performance of local estimator 1. (**a**) $p_{x_{k}}$. (**b**) $p_{y_{k}}$.

**Figure 5 sensors-24-02882-f005:**
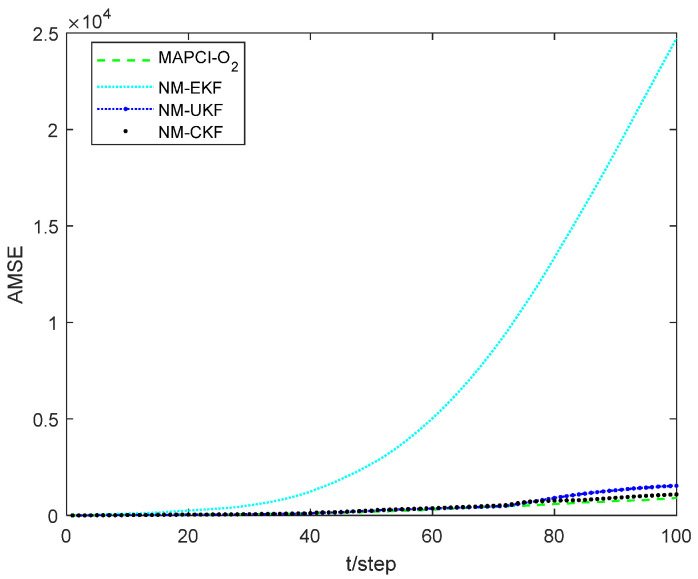
Comparison of AMSE with local estimator 1.

**Figure 6 sensors-24-02882-f006:**
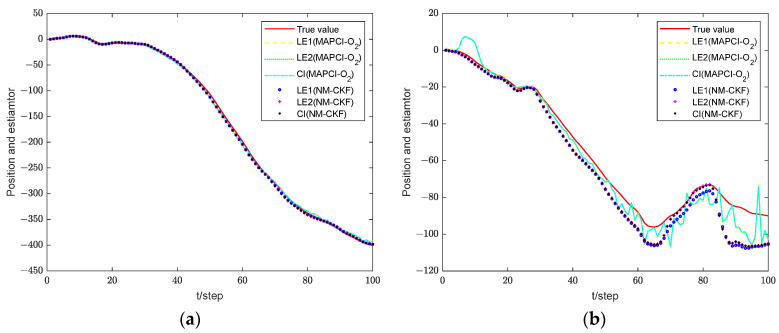
Estimation effect of local and CI fusion estimation algorithms. (**a**) $p_{x_{k}}$. (**b**) $p_{y_{k}}$.

**Figure 7 sensors-24-02882-f007:**
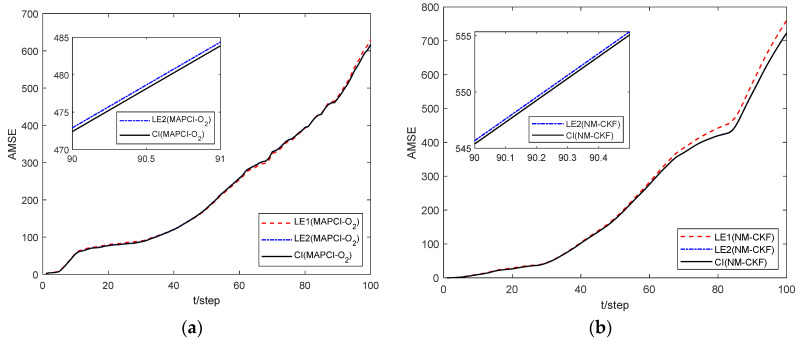
Comparison of estimation performance. (**a**) MAPCI-O_2_ algorithm. (**b**) NM-CKF algorithm.

**Figure 8 sensors-24-02882-f008:**
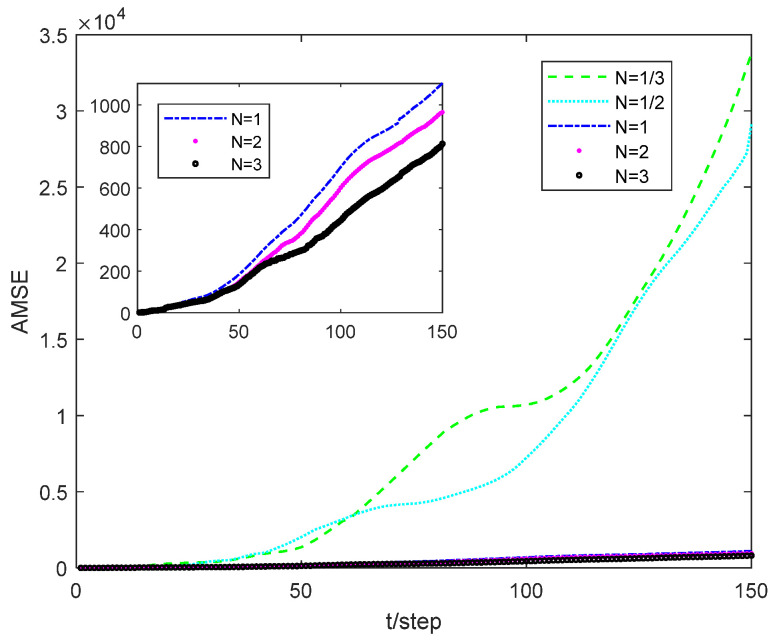
Comparison of AMSE of local estimator 1 with different N.

**Figure 9 sensors-24-02882-f009:**
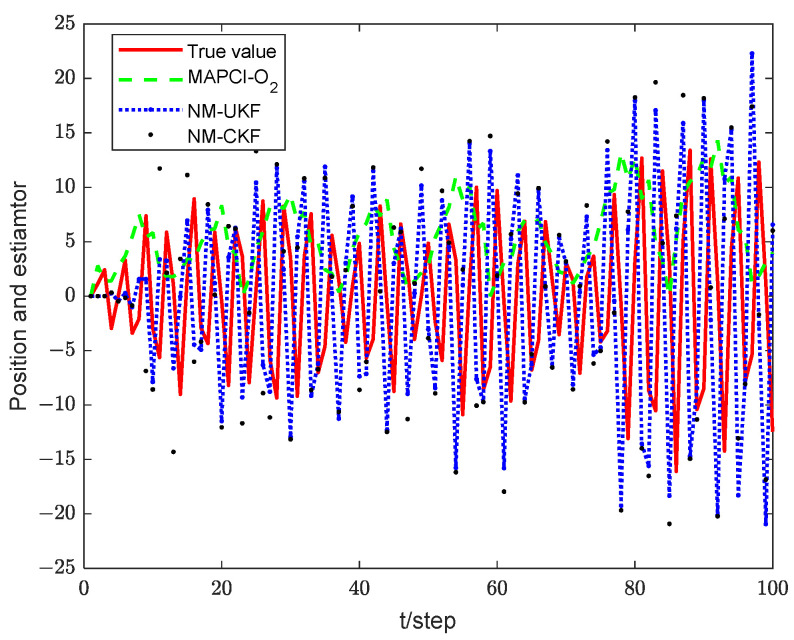
Tracking comparison for different estimators.

**Figure 10 sensors-24-02882-f010:**
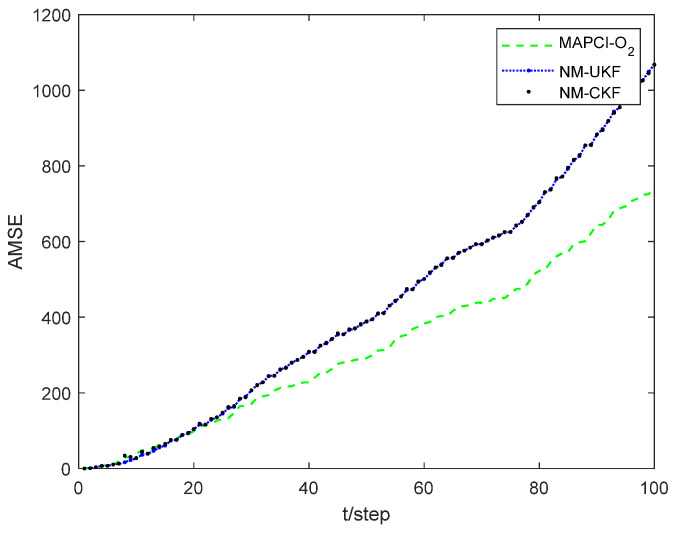
Tracking result for different estimators.

**Figure 11 sensors-24-02882-f011:**
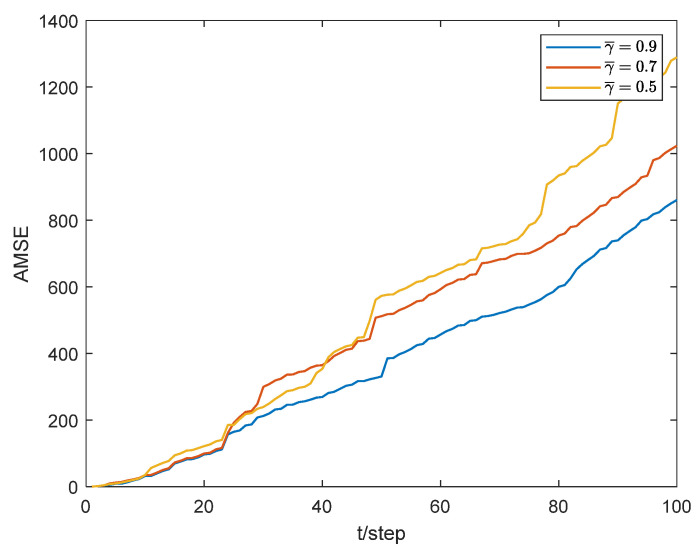
Estimation performance of filters with different packet loss rates.

**Figure 12 sensors-24-02882-f012:**
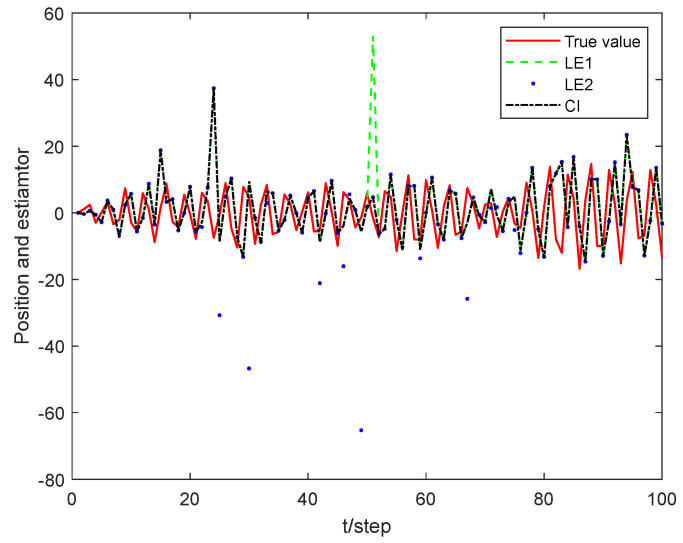
Tracking performance of different estimators.

**Figure 13 sensors-24-02882-f013:**
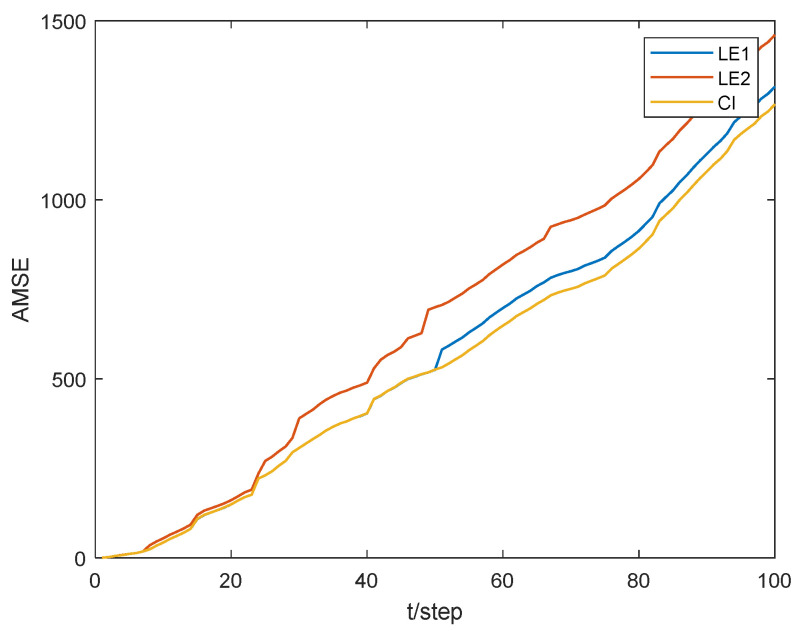
Estimation performance of local and fusion estimators.

**Table 1 sensors-24-02882-t001:** Performance of different estimators.

	AMSE	Computing Time
NM-EKF	6374.7384	0.0067
NM-UKF	433.0898	0.0550
NM-CKF	364.2015	0.0218
MAPCI-O_2_	297.1030	0.0003

**Table 2 sensors-24-02882-t002:** Performance of fusion estimator and local estimators.

	AMSE	
	MAPCI-O_2_	NM-CKF
Local Estimator (sensor 1)	230.1666	245.0515
Local Estimator (sensor 2)	229.7662	234.8488
Fusion Estimator	229.2951	234.7879

**Table 3 sensors-24-02882-t003:** Performance comparison for different estimators.

	AMSE	Computing Time
NM-UKF	428.7306	0.0493
NM-CKF	429.4970	0.0482
MAPCI-O_2_	321.7091	0.0018

## Data Availability

Data are contained within the article.

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
