# Peer review of "A Recursive Non-Uniform Sampling Estimator for Asynchronous Nonlinear Systems"

_sensors, 2024, doi:10.3390/s24092882_

Round 1

Reviewer 1 Report

Comments and Suggestions for Authors

The idea in this paper seems nice, but some points in the paper are not very clear. Some comments and suggestions are as follows:

1. The author should emphasize the contributions of this work at the end of the introduction section more clearly. What are the advantages of the new results? It is still unclear for readers.

2.   It would be good to have an overview of the proposed framework (preferably a system diagram or flowchart), to give readers a quick understanding of the framework.  

3.    The section of Introduction is too long, may be an additional section on Notations is better.
4.      In section II and III, author should analyze how to set the parameters in the framework. Do they have the “optimal” choice?

5. Authors are encouraged to illustrate the potential applications of their framework on some real datasets, to verify the efficiency.

6. Please improve language presentation carefully.

Comments on the Quality of English Language

The idea in this paper seems nice, but some points in the paper are not very clear. Some comments and suggestions are as follows:

1. The author should emphasize the contributions of this work at the end of the introduction section more clearly. What are the advantages of the new results? It is still unclear for readers.

2.   It would be good to have an overview of the proposed framework (preferably a system diagram or flowchart), to give readers a quick understanding of the framework.  

3.    The section of Introduction is too long, may be an additional section on Notations is better.
4.      In section II and III, author should analyze how to set the parameters in the framework. Do they have the “optimal” choice?

5. Authors are encouraged to illustrate the potential applications of their framework on some real datasets, to verify the efficiency.

6. Please improve language presentation carefully.

Author Response

On behalf of all the contributing authors, I would like to express our sincere appreciations of your letter and reviewers’ constructive comments concerning our article entitled “A recursive non-uniform sampling estimator for asynchronous nonlinear systems” (sensors-2927933). These comments are all valuable and helpful for improving our article. According to the reviewers’ comments, we have made extensive modifications to our manuscript and supplemented extra data to make our results convincing. In this revised version, changes to our manuscript were all highlighted within the document by using red-colored text. Point-by-point responses to the nice editors and nice reviewers are listed below this letter.

Reviewer 2 Report

Comments and Suggestions for Authors

In this paper, the research on non-uniform sampling estimator for asynchronous nonlinear systems is meaningful. Suggestions are as follows:

1. The model used in the simulation analysis was too simplistic and the true values were too smooth to reflect the impact of noise on the system. In addition, the effect of the probability of data loss on the tracking performance was not shown in the simulation.

2.  The theoretical analysis of this paper is too brief, and more explanation of the variables in the formula is needed. At the same time, what is the relationship between equations (1), (2) and (7) and (8)?

3. Solving the state estimation in the case of non-uniform sampling is the main problem to be solved in this paper, and there is nothing special about it if it is just a simple interpolation method.

Author Response

(The authors gave the same response as above.)

Reviewer 3 Report

Comments and Suggestions for Authors

This paper investigates the problem of asynchronous estimation of the randomly sampling nonlinear system with packet losses. It seems interesting and reasonable, some minor revisions are needed, i.e.,

1)The vector and matrix should be bold and italic.

2)What's the difference between the developed estimater and traditional observer?

3)The simulation part compares the results with EKF, EKF and CKF, which is good, but further analysis is needed to compare quantitative indicators of these approaches.

4)The main contributions should be further justified, with the new idea or core technology to make the proposed approach novel.

5)Packet losses are reasonable in practice, other scholars often use probit analysis to solve tis problem, e.g., the intermediate observer in 'Spacecraft Attitude Control: A Linear Matrix Inequality Approach' is effective to solve packet losses in parameter estimation, can the authors give some comparison and analysis for this?

6)The whole language should be polished further.

Comments on the Quality of English Language

The whole language should be polished further.

Author Response

(The authors gave the same response as above.)

Round 2

Reviewer 2 Report

Comments and Suggestions for Authors

The author has made serious revisions and enhancements in accordance with the previous recommendations, and according to the new content, the main comments are as follows:

1) The two cases of non-uniform sampling and uniform sampling but random packet loss are different, Figure 2 represents non-uniform sampling, and the probability of packet loss is used in the paper simulation, and the relationship between the two must be explained. 

2) Enhance the simulation analysis of the performance of the distributed fusion estimator.

Author Response

We would like to thank the editors and reviewers for their time and effort in improving the quality of the manuscript. According to these comments, we have made modifications. The following is the detailed response to the comments.

Reviewer 3 Report

Comments and Suggestions for Authors

This paper has been improved significantly.

Comments on the Quality of English Language

none

Author Response

(The authors gave the same response as above.)
